# Exploring Lignin Biosynthesis Genes in Rice: Evolution, Function, and Expression

**DOI:** 10.3390/ijms251810001

**Published:** 2024-09-17

**Authors:** Munsif Ali Shad, Xukai Li, Muhammad Junaid Rao, Zixuan Luo, Xianlong Li, Aamir Ali, Lingqiang Wang

**Affiliations:** 1State Key Laboratory for Conservation and Utilization of Subtropical Agro-Bioresources, Guangxi Key Laboratory of Sugarcane Biology, College of Agriculture, Guangxi University, 100 Daxue Rd., Nanning 530004, China; p2022027@gxu.edu.cn (M.A.S.);; 2College of Life Sciences, Shanxi Agricultural University, Taigu 030801, China; 3Biomass & Bioenergy Research Centre, College of Plant Science & Technology, Huazhong Agricultural University, Wuhan 430070, China; 4State Key Laboratory of Subtropical Silviculture, College of Forestry and Biotechnology, Zhejiang A & F University, Hangzhou 311300, China; 5College of Agriculture, Shanxi Agricultural University, Taigu 030801, China

**Keywords:** rice, lignin monomers, gene families, cell wall, gene expression

## Abstract

Lignin is nature’s second most abundant vascular plant biopolymer, playing significant roles in mechanical support, water transport, and stress responses. This study identified 90 lignin biosynthesis genes in rice based on phylogeny and motif constitution, and they belong to *PAL*, *C4H*, *4CL*, *HCT*, *C3H*, *CCoAOMT*, *CCR*, *F5H*, *COMT*, and *CAD* families. Duplication events contributed largely to the expansion of these gene families, such as *PAL*, *CCoAOMT*, *CCR*, and *CAD* families, mainly attributed to tandem and segmental duplication. Microarray data of 33 tissue samples covering the entire life cycle of rice suggested fairly high *PAL*, *HCT*, *C3H*, *CCoAOMT*, *CCR*, *COMT*, and *CAD* gene expressions and rather variable *C4H*, *4CL*, and *F5H* expressions. Some members of lignin-related genes (*OsCCRL11*, *OsHCT1*/*2*/*5*, *OsCCoAOMT1*/*3*/*5*, *OsCOMT*, *OsC3H*, *OsCAD2,* and *OsPAL1*/*6*) were expressed in all tissues examined. The expression patterns of lignin-related genes can be divided into two major groups with eight subgroups, each showing a distinct co-expression in tissues representing typically primary and secondary cell wall constitutions. Some lignin-related genes were strongly co-expressed in tissues typical of secondary cell walls. Combined HPLC analysis showed increased lignin monomer (H, G, and S) contents from young to old growth stages in five genotypes. Based on 90 genes’ microarray data, 27 genes were selected for qRT-PCR gene expression analysis. Four genes (*OsPAL9*, *OsCAD8C*, *OsCCR8*, and *OsCOMTL4*) were significantly negatively correlated with lignin monomers. Furthermore, eleven genes were co-expressed in certain genotypes during secondary growth stages. Among them, six genes (*OsC3H*, *OsCAD2*, *OsCCR2*, *OsCOMT*, *OsPAL2*, and *OsPAL8*) were overlapped with microarray gene expressions, highlighting their importance in lignin biosynthesis.

## 1. Introduction

Lignin is the second most abundant terrestrial organic polymer after cellulose in plant cell walls, accounting for up to 30% of all vascular plant tissue [1]. Lignin is hydrophobic compared to hydrophilic polysaccharide components of plant cell walls and plays a significant part in conducting water in plant stems. The crosslinking of polysaccharides by lignin is an obstacle to water absorption onto the cell wall. Therefore, lignin allows the plant’s vascular tissue to conduct water efficiently [2]. Lignin exists in almost all vascular plants, except bryophytes, supporting the idea that the original function of lignin was restricted to water transport [3]. Lignification provides large upright vascular plant forms, which enable some species to more successfully compete for photosynthetic energy [4]. This, in turn, provides the molecular or structural basis for much of the plant biodiversity that humanity enjoys today in its many splendid forms. Furthermore, in addition to competition for light, an upright growth habit allows better spore/pollen dispersal, increasing the genetic variability and species range. Our knowledge of plant cell wall assembly is at the most rudimentary level, till now [5].

The phenylpropanoid metabolic pathway produces lignin through three steps [1,2,6]. During the initial stage of photosynthesis, known as the shikimate pathway, plants convert glucose into aromatic amino acids, like tryptophan, phenylalanine, and tyrosine. The second stage involves the conversion of phenylalanine into associated products by enzymes, such as phenylalanine ammonia-lyase (PAL), cinnamate 4-hydroxylase (C4H), coumarate 3-hydroxylase (C3H), shikimate hydroxycinnamoyl transferase (HCT), caffeic acid O-methyltransferase (COMT), caffeoyl CoA O-methyltransferase (CCoAOMT), and ferulate 5-hydroxylase (F5H). Lastly, 4-coumarate CoA ligase (4CL) produces the corresponding coenzyme A thioesters. Cinnamoyl CoA reductase (CCR) and cinnamyl alcohol dehydrogenase (CAD) transform the middle-stage products into three lignin monomers in the third stage. The three lignin monomers are syringyl lignin (S-lignin), p-hydroxyphenyl lignin (H-lignin), and guaiacyl lignin (G-lignin) [7]. The lignin composition differs across various plant groups; dicots comprise G-lignin and S-lignin, while monocots contain all three types, though H-lignin tends to be in lower amounts. In contrast, gymnosperms and ferns are mainly characterized by G-lignin [8].

Genome-wide surveys based on sequence classification and annotation have identified 34, 56, 35, 104, 117, and 37 lignin synthesis genes in *Arabidopsis* [9], *Setaria viridis* [10], *Pyrus bretschneideri* [11], banana [12], maize [13], and *Eucalyptus grandis* [14], respectively. Rice is the most important grain in human nutrition and caloric intake, providing more than one-fifth of the calories consumed worldwide by the human species [15]. It is also emphasized as a model species for the functional genomic characterization of monocotyledon plants [16]. With the completion of the rice genome sequence, lignin-related gene families have been identified in rice [17]. It has shown a striking difference in lignin-related gene families between rice and *Arabidopsis*, reflecting dicots and monocots’ distinct cell wall compositions [18]. On the other hand, several orthologs of *Arabidopsis* lignin-related genes have shown a similar function in rice [19]. Research across species has identified genes responsible for specific lignin monomers or total lignins. For example, reduced expression of the *COMT* gene led to a notable decrease in S-lignin levels in transgenic switchgrass [20] and sorghum bicolor [21]. Among *OsCADs, OsCAD8A* and *8C* exhibited the second-highest appearance in the rice lignin biosynthesis co-expression network [22]. *OsCAD2* was suggested to be the sole rice gene belonging to the bona fide CAD lineage, by a previous phylogenetic analysis. However, knockout mutants of *OsCAD2* exhibited a slight decline in Klason lignin quantity (5–6 percent), suggesting other *OsCADs* may also play important roles in lignin formation [23]. Knockout mutants of *OsCAD2* in another study were also reported to contain slightly reduced lignin contents but significantly increased G-lignin (16%) and highly increased H-lignin (34%), which increased biomass saccharification yields through enhancing hexoses production by 61–72% without significant loss of biomass [24]. Contrarily, a forty percent reduction in Klason lignin content was observed in *Arabidopsis* double mutants (*atcad-c*/*-d*), indicating the contribution of CADs to lignin contents may vary among rice and *Arabidopsis* [23]. Additionally, *fc1*, a T-DNA-tagged *OsCAD7* mutant, exhibits a 34% decrease in mechanical strength at the topmost internode and 13.4% less lignin [25]. *Os4CL3* and *Os4CL5* have been known to be activated by *OsMYB30*, resulting in increased lignin production in sclerenchyma cells to prevent *M. oryzae* infection [26]. *OsPAL1/2/5*, *Os4CL1/3/4*, and *OsCAD2* are co-expressed with *OsMYB110*/*30*/*55* and several lignin metabolites, suggesting *OsMYBs* regulated lignin biosynthesis [27]. The association between *Os4CL3* and G-lignin was found in knock-out mutants of *Os4CL3,* exhibiting significantly reduced G-lignin monomers [28]. Additionally, overexpression lines of *F5H* lead to very high S-monomer contents in hybrid poplar [29].

Although the lignin monomer biosynthesis grids are well elaborated, due to the complexity of the pathway, the genes related to individual lignin monomers and total lignin contents are still unknown, which hampers the deep study of lignin production pathways in rice. Furthermore, earlier research has focused on individual lignin genes or gene families in rice. Therefore, a combined analysis of multi-gene families was needed. We identified ten gene families linked to lignin biosynthesis through homology-based search methods. A detailed analysis was conducted on the protein evolutionary relationships, protein domains, and expression patterns of these genes. For an in-depth analysis of the lignin biosynthesis pathway genes, we selected five rice genotypes that exhibited significant variations in stem strength. Lignin monomers and gene expression measurements in the stems of these genotypes at five different growth stages led us to identify the putative candidate genes related to lignin biosynthesis. The outcomes of this research not only enhance our understanding of lignin formation genes at the genomic level in rice and identify several candidate genes, they lay the groundwork for further exploration of the molecular mechanisms governing lignin formation and regulation. Furthermore, this research provides important comprehensions that can be applied in other plant species to study lignin production and lodging tolerance.

## 2. Results

### 2.1. Identification and Evolutionary Relationships of Potential Lignin Biosynthesis Genes in the Rice Genome

In this study, we identified 90 potential *lignin monomers biosynthesis* genes through searches in the rice genome, and they belonged to phenylalanine ammonia-lyase (PAL, 9), cytochrome p450 (P450s, 8), 4-hydroxycinnamoyl CoA ligase (4CL, 14), hydroxycinnamoyl-CoA: shikimate/quinate hydroxycinnamoyltransferase (HCT, 8) C3H, caffeic acid O-methyltransferases (COMTs, 8), caffeoyl CoA O-methyltransferases (CCoAOMTs, 6), cinnamoyl CoA reductase (CCR, 26), F5H, and cinnamyl alcohol dehydrogenases (CADs, 11) families (Table 1). A search of the CDD database shows 90 significantly matched sequences of lignin-related genes in rice (Appendix A and Figure 1). The phylogenetic analysis indicated that OsPALs, OsCHs, and OsCADs were grouped into three distinct categories, whereas Os4CLs were classified into four groups. In contrast, OsCCoAOMTs, OsHCTs, and OsCCRs were distributed across two clades. Furthermore, domain analysis of OsPALs suggested that the rest of the proteins had dimerization domains except for five proteins. Surprisingly, other gene families’ proteins comprised single characteristic domains.

The 90 genes were mapped to rice chromosomes (Figure 2). Chromosomes 2, 8, and 9 exhibited the highest abundance of lignin-related genes, with 17, 14, and 13 genes, respectively. Conversely, chromosomes 7, 11, and 12 possessed two, three, and three genes, suggesting an uneven gene distribution in the rice genome. The gene duplication analysis identified three pairs, three triplets, three quadruples of tandem duplicated genes, and nine pairs of segmentally duplicated genes. Furthermore, chromosomes 2, 4, 6, 8, and 9 were highly enriched with duplicated genes compared to others.

The members of the lignin biosynthesis gene families exhibited the conserved and specific features in their gene structures and protein motifs (Figure 3). *OsCHs* and *OsF5Hs* possessed two to four exons, while motif 30 was found conserved among all proteins of this family except OsC3H, which only possessed motif 9 (Figure 3A). Among *OsCCoMTs, OsCCoMT6* was an exception compared to other members of this gene family as it contained nine exons, while the rest had three to five exons (Figure 3B). The differences were also evident in conserved protein motifs as they contained the specific motif 6 and conserved motif 24, which was present among the whole family. In the *OsCAD* gene family, most members encode proteins by four, five, or six exons, while the duplicated genes of the *OsCAD8A*/*B*/*C*/*D* subgroup only contained two exons (Figure 3C). The motif analysis exhibited five motifs (motifs 2, 21, 26, 3, 18) that were conserved amongst all OsCADs, except OsCAD1, which lacked motif2. Most *OsCCRs* contained five exons, while some members had three, four, or six exons, except *OsCCR9*, which was intronless (Figure 3D). Similarly, four motifs (7, 10, 12, 13) were among all OsCCRs with a few exceptions, such as OsCCRL1 had only motif 7, OsCCRL6 lacked motif 7, and OsCCR1 lacked motifs 10 and 12. These results suggest motifs 7 and 13 might be necessary for the functions of the CCR gene family in rice. The gene family members of *Os4CLs* contained two to six exons and predominantly the first or last exons were considerably larger in size (Figure 3E). Four protein motifs (4, 14, 15, 20) were conserved among most Os4CLs. Os4CLs can be partitioned into two groups based on additional motifs, along with four conserved motifs. Group I possessed the specific motif 8, while Group II contained additional motifs 25 and 29. Genes comprised a relatively lesser number (two, three, or four) of exons in *OsCOMTs*. Four protein motifs (9, 17, 22, and 27) were common among most of the COMTs proteins analyzed. OsCOMT and OsCOMTL7 contained an additional motif 15, whereas OsCOMTL6 had an additional motif 19 similar to the majority of Arabidopsis AtCOMTLs. OsCOMTL2/5 only comprised motif 17, while OsCOMTL4 consisted of merely motifs 12 and 17. The results suggest motif 17 was a conserved motif for this protein family. *OsPALs* consisted of a single or two exons, while seven motifs (6, 16, 5, 1, 23, 28, 11) were conserved among most OsPALs (Figure 3F). OsPAL2 only contained motifs 6, 11, and 16, while OsPAL3 lacked motif 28 among seven conserved motifs. Like *OsPALs*, *OsHCTs* also contained one or two exons, though their encoded proteins lacked common conserved motifs (Figure 3G). These results indicate the diversity and commonalities in the structural organization of lignin biosynthesis genes and proteins. 

In multiple sequence alignment analysis, we performed a comparison between rice lignin biosynthesis proteins conserved domains with *Arabidopsis*. We discovered that every protein sequence has the core conserved domain or motifs that are associated with the characteristics of the gene family. Eight conserved motifs usually occur in the protein sequence of the CCoAOMT gene family, and among them, motifs D, E, F, G, and H are specific to the CCoAOMT family and function as tag sequences [30]. Our results indicate that motif F (AGVAHK) and motif G (FAFVDADK) are conserved among all the OsCCoAOMTs analyzed (Figure 4A). All the OsPALs possessed the highly conservative motif (GTITASGDLUPLSYIAG) (Figure 4B). The heme-binding conserved amino acid motif (PFGSGRRSCP) and proline-rich PERF motif of cytochrome P450 conservative domains were present in all the OsC3Hs, OsC4Hs, and OsF5Hs (Figure 4C). Conserved Box I (SSGTTGLPKGV) was present in all Os4CLs; however, Box II was absent (Figure 4D), contrary to a previous sequence analysis in Ma4CLs [12]. Acetyltransferase gene family conserved motifs (DFGWGR) and active motifs (H3XDG) were present in all OsHCTs (Figure 4E). Partial homology was observed for the CCR family conserved motif (KNWYCYGK) among OsCCRs (Figure 4F). Four conserved amino acid motifs (motif 1: LVDGGGxG, motif II: GINFDLPHV, motif III: EHVGGDMF, and motif IV: NGKVI) have been found in the COMT protein sequences [31]. However, sequence alignment analysis indicated motifs I and II were completely conserved, while motif IV was completely absent (Figure 4G). Interestingly, OsCOMTL2/4/5 lacked motif III completely, whereas the rest of the members retained it, suggesting it might have led to functional divergence. Characteristic motifs of the CAD family, such as the Zn^2+^ binding box (GHE.2X.G.5X.G.2X.V), NADPH binding (G.x.2G.VG), and Zinc binding box (C.xx.C.xx.C.x7.C), were conserved among the majority of OsCADs (Figure 4H). The results indicate that all 90 rice lignin-related genes possess characteristic conserved domains specific to their respective gene families.

### 2.2. Co-Expression Profiling of Lignin Biosynthesis Genes in Rice

Hierarchical clustering was utilized to categorize the transcript abundance patterns of the potential lignin synthesis-related genes. This method allowed us to combine an additional criterion for narrowing down the list of candidate genes. The gene expressions of lignin biosynthesis genes could be divided into two large clusters (Figure 5). Cluster I comprised low-expressed genes, while Cluster II contained highly expressed genes. *Os4CL4*, OsC4H1, *OsCAD8B*, *OsCCR6*/*7*/*10*/*11*/*12*, *OsCCRL10*, *OsCOMTL2*, *OsCOMTL2*/*4*/*6*, *OsF5H3*, *OsPAL4*, and *OsHCT4*/*8* (cluster I, Figure 5) exhibited extensively low expressions in most tissues examined. Further grouping within Cluster II exhibited that lignin-related genes within IIA might be involved in primary cell wall synthesis, as they were co-expressed with primary cell wall-related genes. The member genes *OsCCRL11*, *OsHCT1*, *OsCCoAOMT5*, *OsCOMT*, *OsC3H*, *OsCAD2*, *OsPAL1*, and *OsPAL6* genes showed extremely high expression over all tissues of life cycles, suggesting their constitutive gene expressions. Similar trends for cluster IIA genes were also observed in Nipponabare (*Japonica*) gene expressions (Appendix A), suggesting their co-expression is conserved across both *indica* and *japonica* accessions. On the other hand, 15 IIC subclass genes (*Os4H4*, *OsHCT7*, *OsCCRL1*/*3*/*4*/*7*/*9*, *OsF5H1*, *OsCAD3*/*8C*, *OsCCR1*/*5*, *OsPAL5*/*8*/*9*) might play a role in secondary cell wall formation as they are co-expressed with the secondary cell wall-specific marker genes. In summary, lignin-related gene expression levels were the highest in the stem, hull, and root, and were relatively low in the panicles, seeds, and stamen.

Earlier research has reported various genes, such as *AtCAD1* [32], *AtPAL1*/*2*/*4* [9], *AtF5H* [33], and numerous other genes, are involved in lignin biosynthesis. The identification of orthologous genes in *Arabidopsis* using rice lignin-related genes as queries followed by hierarchical clustering analysis of RNA-seq expression profiles provided additional valuable insights. The analysis revealed three main categories in the 63 *Arabidopsis* lignin gene expression profiles (Appendix A). Additionally, we provided an overview of co-expression patterns in both rice and *Arabidopsis* for lignin synthesis pathway genes (Table 1). The expression profiles of the genes belonging to the families associated with *Arabidopsis* lignin synthesis exhibited similarities and differences with the rice. For example, *Os4CL3*, *OsC3H*, *OsCCoAOMT1/5*, *OsHCT8*, and *OsPAL1*/*2*/*6*/*8*, preferentially expressed in panicles, had orthologs *At4CL1*, *2*, and *4*, *AtC3H1*, *AtCAD1*, *AtCCoAOMT3*, and *AtPAL1*, *2*, and *4*, which were highly expressed in siliques and seeds. As an instance of genes exhibiting varied expression files, there were no genes exhibiting an exceptionally low expression level, such as those found in cluster I (Figure 5) in rice, suggesting a higher expressional redundancy of lignin synthesis genes in rice compared to *Arabidopsis*. Taken together, the pattern of gene co-expression may represent both the resemblances and alterations between rice and the *Arabidopsis* plant’s cell wall compositions.

The expression profile of the seventeen sets of lignin-related duplication genes (nine tandem duplication and eight segmental duplication sets) with the corresponding probes was analyzed. We found a divergent expression pattern within a duplicated set (Figure 6). Of the seventeen gene sets, only *OsPAL6* and *OsPAL8* in tandem duplicates exhibited relatively similar expression patterns. The fate of three pairs (*OsCCR9*/*10*/*11*, *OsCAD8A*/*B*/*C,* and *Os4CL3*/*4*) could be described as nonfunctionalization, where one member of the set lost expression in all tissues, while the other showed strong expressions. In the other duplication sets, the expression patterns of both member genes were partially complementary or overlapped. Comparison of expression pattern shifts of the duplicated genes of the lignin-related genes could reflect the divergence hypotheses that a duplicate gene pair might be involved in nonfunctionalization, subfunctionalization, and neofunctionalization.

### 2.3. Biochemical Analysis of the Lignin Monomers in Rice Genotypes

The amount of lignin and the makeup of its monomers are crucial factors in the biomass conversion and saccharification processes [34]. Additionally, lignin content and monomer composition change throughout different stages of rice growth. Therefore, we determined the lignin monomer composition in five genotypes, such as Nipponbare (NPB), C6 (T-DNA mutant in NPB/fragile culm *Osfc9*/*OsMYB103L*), C15 (*Osfc16* mutant/*OsCESA9*), C17 (brittle culm3/bc3 mutants), and Y102 (knock down mutants of *Os4CL4*)), at six different growth stages (Figure 7 and Appendix A). The total lignin, along with monomer contents, exhibited increasing trends with the progression of growth. 

We further calculated the growth value of the monomer through the difference between the previous period and the next period. However, the monomer increments between consecutive stages were not uniform (Appendix A). Except for C6, four genotypes transition from stage five to six exhibited the highest increase in lignin monomer contents. However, the increase was significant for all the stages.

### 2.4. RT-qPCR Assays of Twenty-Seven Selected Genes and Their Association with Lignin Monomer Synthesis

It was assumed that significant increases in monomer content would correlate with notable differences in the expression of genes regulating their production. Consequently, we selected twenty-seven genes (Appendix A) highly expressed in old stems, panicles, and hull tissues related to secondary cell wall synthesis and lignification in the CREP microarray data. The qPCR assays of the selected twenty-seven genes were performed using the same growth stages and genetic materials, as in the prior biochemical analysis of lignin monomers.

The correlation analysis identified significant negative associations between gene expressions and lignin monomer contents (Figure 8). Among twenty-seven genes, *OsPAL9* exhibited highly significant negative correlations for all lignin monomers, while *OsCAD8C* showed significant negative correlations for the S-monomer in the C6 genotype (Figure 8A). Interestingly, *OsCCR8* exhibited highly negative associations with H and S-monomers in the C17 (Figure 8B) and with all types of monomers in the Y102 mutants (Figure 8C), indicating *OsCCR8* roles in different genotypes. Similarly, *OsCOMTL4* was highly negatively correlated with S-lignin monomers in the Y102 genotypes, suggesting some lignin-related genes might be genotype-specific. Surprisingly, no significant associations of gene expression with monomer contents were identified in NPB and C15 genotypes. The analysis suggested negative correlations between the four genes and monomer contents. The five genotypes’ correspondence between lignin monomers and gene expressions (qPCR results) was variable. 

### 2.5. Identification of Common Promising Genes among CREP Array and qPCR Expressions

Co-expression explains how similar patterns of gene expression may perform or regulate similar functional pathways [35]. The relationship between gene expression and genotype growth stages is depicted in Figure 9. Based on qPCR results, eleven genes (*OsCAD1*, *OsCCoAOMT1*, *OsC4H2*, *OsCCoAOMT5*, *OsCCR2*, *OsC3H*, *OsPAL8*, *OsCAD2*, *OsCOMT*, *OsPAL2*, and *OsHCT1*) on the lignin synthesis pathway were grouped according to their co-expressions in certain tissues and genotypes (Figure 9A). Furthermore, nine genes (*Os4CL1*, *Os4CL3*, *HCT2*, *OsC3H*, *OsCAD2*, *OsCCR2*, *OsCOMT*, *OsPAL2,* and *OsPAL8*) were found to be highly expressed in the old panicle, stem, and hull tissues from the CREP microarray data (Figure 9B). Finally, six genes (*OsC3H*, *OsCAD2*, *OsCCR2*, *OsCOMT*, *OsPAL2,* and *OsPAL8*) were found to be common between both the datasets (Figure 9C), which we herein dubbed as core genes for lignin biosynthesis. 

## 3. Discussions

Lignin, a complex organic polymer, is a crucial element of plant cell walls, and it plays vital functions in sustaining normal growth, improving the overall mechanical strength, and boosting the stress tolerance of plants [36]. The biosynthesis of lignin is a complex process involving a series of enzymatic reactions that convert phenylpropanoid precursors into the lignin polymer. The identification and characterization of the genes encoding these enzymes are important for advancing our understanding of lignin biosynthesis, and for producing optimal new varieties that exhibit great resistance and superior quality. This study highlights the importance of integrating biochemical analysis with gene expression data to better understand the lignin biosynthetic pathway.

In the present research, we identified 90 genes from the rice genome across ten gene families involved in lignin production (Figure 1). The phylogenetic divergence of nine OsPALs into three groups was similar to the PALs reported in other studies [37]. We identified 14 Os4 CLs distributed into two types of seven proteins each. Type I is mainly involved in lignin formation, whereas type II is often linked with flavonoid biogenesis [38]. Our analysis suggests rice contains an equal number of *4CLs* for lignin and flavonoid biosynthesis. Variable numbers of COMTs have been identified in plant species. For instance, 18, 27, 25, 7, and 24 *COMTs* have been reported in *Arabidopsis*, grape vines [39], *Populus trichocarpa* [37], Eucalyptus [14], and banana [12], respectively. Eight *OsCOMTs* were identified in this research. *OsCOMT1* and *AtOMT1* were reported to be involved in lignin formation in rice and *Arabidopsis*, respectively [40,41,42]. Such results suggest other uncharacterized *OsCOMTs* could also be involved in lignin synthesis. The CAD gene family can be classified into three types based on substrate preference and homology [10]. Functional studies of *AtCAD1*/*4*/*5* and *OsCAD2* have suggested their involvement in lignin biogenesis [22,43]. Our analysis revealed that OsCAD1/4 shares high homology with AtCAD1, suggesting a significant role in lignin production. Downregulation of *AtCCR1* in Arabidopsis results in a 50% reduction in lignin content and weakened secondary cell walls [44]. The phylogenetic analysis indicated a strong homology between AtCCR1 and OsCCR1/2/3/4/5/6, suggesting these rice orthologs are crucial for lignin biosynthesis. 

Gene duplications are crucial for genome evolution and chromosomal reorganization following genome-scale duplication events [45]. The rice genome underwent a single whole genome duplication and conserved segmental duplication events [46]. Our analysis identified eight segmentally duplicated and nine tandemly duplicated gene groups within rice lignin gene families, indicating the significant role of duplications in rice genome evolution (Figure 2). These results suggested segmental and tandem duplication events contributed largely to the expansion of lignin-related gene families.

Lignin plays a significant role in the cell walls of vascular plants, where it primarily accumulates in the secondary walls of vascular, mechanical, and protective tissues. Therefore, we used primary (*OsCESA1*/*3*/*8*) and secondary (*OsCESA4*/*7*/*9*) cell wall marker gene [47] expressions to classify lignin biosynthesis gene expressions (Figure 5). Our analysis showed that *OsCOMT*, *OsPAL1*/*6*, *OsCAD2*, *OsCCoAMT5*, *OsC3H*, *OsCRL11*, and *OsHCT1* members of the IIA cluster co-expressed with primary cell wall markers. On the other hand, *OsCAD3*/*8*, *OsCCR3*, *OsC4H4*, *OsHCT7*, *OsCCR1*/*4*/*5*/*7*/*9*, *OsF5H1*, *OsPAL5*/*8*/*9* members of the subcluster IIC co-expressed with secondary cell wall markers (Figure 5). Lignin-related genes are strongly expressed in the old and young stems, radicle, hull, and spikelets tissues, indicating their significant involvement in producing lignin.

We observed a gradual increase in the lignin content of the rice stem across different growth stages in five cultivars (Figure 7), consistent with a previous study on rice. This rise in lignin content may be attributed to the lignification process [48]. To understand the relationship between lignin gene expressions and lignin composition, twenty-seven highly expressed genes from a microarray dataset were confirmed via RT-qPCR. Correlation analysis revealed significant negative relationships between gene expressions and lignin monomer contents (Figure 8). *OsPAL9* expression was negatively correlated with S, H, G, and total lignin in C6 (Figure 8A). To the best of our knowledge, there are no reports on *PAL’s* effects on individual lignin monomers; however, overexpression of *OsPAL8* resulted in higher lignin content [49]. Similarly, *OsCAD8C mRNA* transcripts were negatively associated with S-lignin in C6, in conformation to a study reporting a novel aspen gene encoding PtSAD, a homologous protein to PtCAD, is essential for forming S-type lignin [50]. Meanwhile, S and H-monomers showed similar negative relationships with *OsCCR8* expression in C17 (Figure 8B). Furthermore, *OsCCR8* demonstrated significant negative correlations with S, H, and G-monomers and total lignin in the Y102 genotype (Figure 8C). Downregulation of *OsCCR* gene expression in rice exhibited reduced lignin accumulation in both the anthers and roots [51]. It still needs to be confirmed whether *OsCCR8* and *OsCAD8* affect individual monomers or total lignin content through further research. Currently, it is known that modifying *CAD* and *CCR* genes reduces total lignin content rather than affecting specific monomers [14,52,53]. Lastly, *OsCOMTL4* negatively correlated with S-lignin in Y102 (Figure 8C). COMT is crucial in determining lignin composition by methylating 5-hydroxypinobanksyl aldehyde, 5-hydroxypinobanksyl alcohol, and caffeic acid, which contributes to the production of mustard aldehyde, mustard alcohol, and ferulic acid, respectively. These reactions are essential for the methylation of pinobanksyl alcohol (G-lignin) and mustard alcohol (S-monomer) [7,54]. 

Earlier studies in rice [48] and maize [13] suggested lignin biosynthesis-related gene expressions are increased at the later growth stage after stem morphogenesis. This indicated that enhanced lignin concentration and stem strength were caused by increasing expression of genes involved in lignin production in the stem [13]. Our previous analysis indicated that only four genes have linear relationships to increasing lignin monomers or total lignins (Figure 8). Therefore, we employed a co-expression approach to identify gene expressions enriched at specific growth stages. Eleven genes showed highly similar co-expression patterns (Figure 9A). These genes were lowly expressed in the early and late stages of stem growth but highly expressed during the booting and heading stages (C6-III, Y102-III, and C17-III), which are crucial for lignification and stem development. The enrichment of gene expression at Stage III aligns with the rapid increase in lignin content at this stage compared to Stage II (Figure 9A). Previous rice studies observed a significant increase in lignin from Stages II to III and IV to V, with a gradual rise over five growth stages [48]. However, the core lignin genes, such as *OsCAD2*, and *OsCOMT*, along with others, expression declined after peaking at Stage III in this study, suggesting negative feedback mechanisms might have switched off key lignin biosynthesis genes as plants transitioned from vegetative to reproductive phases. In contrast, lignin content continued to rise until maturity. Further research is required to understand these complex relationships. Among the eleven candidate genes analyzed for RT-qPCR differential co-expression (Figure 9A), six showed expression patterns consistent with their microarray expression profiles (Figure 9B,C), indicating higher transcript accumulation in various tissues of secondary growth. Notably, two genes, *OsCAD2* and *OsCOMT*, exhibited high expression in lignin-synthesizing tissues in GUS assays [22], validating the effectiveness of our analysis.

This study identified correlations between lignin content and the expression of lignin synthesis genes. Increased gene expression in stems may lead to higher lignin content and enhanced stem strength, likely due to lignin polymerization’s role in forming thick-walled vascular bundle cells. Future research can target the six candidate lignin biosynthesis genes and four genes associated with lignin monomers identified here, and the methodology used can be applied to other crops. However, the actual function of the gene requires further experimental validation, such as gene over-expression, gene editing, and so on. Studies of the evolution, function, and expression of lignin biosynthesis genes provide an excellent model for studying the coordinated action of biochemical pathways in plants and will guide us in developing better strategies for improving bioenergy feedstocks.

## 4. Materials and Methods

### 4.1. Database Searching for Lignin Biosynthesis Genes in Rice

Hidden Markov Model (HMM) profile of PAL (PF00221), C4H (PF00067), 4CL (PF04443), HCT (PF02458), C3H (PF00067), CCoAOMT (PF01596), CCR (PF01370), F5H (PF00067), COMT (PF00891), CAD (PF08240) domains were downloaded from Pfam (http://pfam.sanger.ac.uk/). Putative function search was carried out for Phe ammonia lyase (PAL), trans-cinnamate 4-hydroxylase (C4H), 4-coumarate: CoA ligase (4CL), hydroxycinnamoyltransferase (HCT), p-coumarate 3-hydroxylase (C3H), caffeoyl-CoA 3-O-methyltransferase (CCoAOMT), cinnamoyl-CoA reductase (CCR), ferulate 5-hydroxylase (F5H), caffeic acid O-methyltransferase (COMT), cinnamyl alcohol dehydrogenase (CAD) genes in the Rice Genome Annotation Project (RGAP 7) database. With BLASTP searches, the homologs for lignin-related proteins in rice were also identified. For *Arabidopsis*, a dataset representing all lignin-related genes encoded by the *Arabidopsis* genome was generated based on a putative function search in the Arabidopsis Information Resource (TAIR) database (http://www.arabidopsis.org/ (accessed on 20 October 2022)).

To gain an understanding of the evolutionary relationship between *Arabidopsis* and rice lignin-related genes and to identify a basis for classifying newly uncovered members, phylogenetic analysis was performed for lignin-related genes from the *Arabidopsis* and rice genomes. Preliminary phylogenetic analysis was conducted to choose and name the closely related lignin related genes from *Arabidopsis*. Information about the chromosomal localization, coding sequence (CDS), amino acid (AA), and full-length cDNA accessions was obtained from RGAP and KOME (http://cdna01.dna.affrc.go.jp/cDNA (accessed on 25 November 2022)). The corresponding protein sequences were confirmed with the Pfam database (http://pfam-legacy.xfam.org/ (accessed on 10 December 2022)). 

### 4.2. Phylogenetic Analyses and Motif Identification

A multiple alignment was performed using the MEGA (v11) software [55]. Several values for gap opening penalty and gap extension penalty were tried to identify the commonly resolved domain. A combination of gap opening penalty = 10.0 and gap extension penalty = 0.2 was finally adopted, which enabled reasonable alignment among conserved domains with a few gaps. Phylogenetic analyses of lignin-related proteins were carried out using the NJ, MP, and maximum likelihood (ML) methods in MEGA, respectively. NJ analyses were conducted with the pairwise deletion option selected and with the Poisson correction set for the distance model. For parsimony analysis, 1000 resamplings for bootstrap tests were performed. Support for each node was tested with bootstrap analysis, 1000 replicates for NJ and 100 for MP and ML. Since the MP and ML trees do not disagree with the NJ trees in topology, we present only the NJ trees in this paper. Further, we display and manipulate phylogenetic trees through the online tool called Interactive Tree Of Life (http://itol.embl.de/ (accessed on 18 December 2022)). 

Protein sequences were analyzed in the PROSITE program (http://prosite.expasy.org/ (accessed on 20 December 2022)) and PRATT program (http://web.expasy.org/pratt/ (accessed on 25 December 2022)) for patterns conserved in protein families, domains, or a set of protein sequences. Protein sequences were analyzed in the Conserved Domain Database (CDD) (http://www.ncbi.nlm.nih.gov/Structure/cdd/wrpsb.cgi (accessed on 28 December 2022)) search program to find conserved domains, with E-value = 0.01 as the cutoff. Gene structure information was drawn from genome GFF3 files, while conserved motifs were identified through the MEME tool (https://meme-suite.org/meme/tools/meme (accessed on 9 January 2023)) and drawn using the TB tool (v2.096) [56]. The protein sequence’s multiple alignments were performed in MEGA software and later visualized through the ESPript 3.0 web tool (https://espript.ibcp.fr/ESPript/ESPript/index.php (accessed on 15 January 2023)).

### 4.3. Chromosomal Localization and Gene Duplication Analysis

Lignin-related genes were mapped on chromosomes by identifying their chromosomal positions given in the RGAP rice database. The duplicated genes were elucidated from the segmental genome duplication of rice through Tbtools (v2.096). Genes separated by five or fewer genes were considered to be tandem duplicates. The distance between these genes on the chromosomes was scaled by the Physical Maps by the Oryzabase Map Tool program (http://viewer.shigen.info/oryzavw/maptool/MapTool.do (accessed on 19 January 2023)).

### 4.4. Genome-Wide Expression Analysis of Lignin-Related Genes in Rice and Arabidopsis

To understand the functions of rice lignin-related genes, we investigated their expression patterns in rice tissues by microarray analysis. The tissues included calli, plumule, radicle, leaf, sheath, flag leaf, panicle, stem, endosperm, root, seed, shoot, hull, spikelet and stamen. Expression profile data of lignin-related genes in 66 tissue examples of *Indica* rice cultivars Zhenshan 97 (ZS97) and Minghui 63 (MH63) were obtained from the CREP database (http://crep.ncpgr.cn (accessed on 24 January 2023)), from the rice transcriptome project using Affymetrix Rice GeneChip microarray. Furthermore, the same 90 rice lignin biosynthesis gene spatiotemporal expression profiles in Nipponbare (*Japonica*) normalized signal intensity plot data were downloaded from the RiceXPro database (https://ricexpro.dna.affrc.go.jp/ (accessed on 27 January 2023)). Massively parallel signature sequencing (MPSS) data (http://mpss.udel.edu/rice/ (accessed on 29 January 2023)) was used to determine the expression profiles of the genes with conflicting probe set signals. By convention, expression values were logarithmized for CREP data, and cluster analyses were performed using the software Tbtools (v2.096) with Euclidean distances and the hierarchical cluster method of “complete linkage clustering”. All microarray data of *Arabidopsis* were downloaded from the Gene Expression Omnibus database (http://www.ncbi.nlm.nih.gov/geo/ (accessed on 1 February 2023)). Subsequent analysis of the gene expression data was performed in statistical computing language R (http://www.rproject.org (accessed on 4 February 2023)) using pheatmap packages available from the Bioconductor project (http://www.bioconductor.org (accessed on 4 February 2023)). 

### 4.5. Cultivation of Plant Materials, Sample Collection and Pretreatments

Five genotypes, including Nipponbare (NPB) as Wild type, while four mutants as comparison groups, such as C6 (T-DNA mutant in NPB/fragile culm *Osfc9*/*OsMYB103L*) [57], C15 (*Osfc16* mutant/*CESA9*) [58], C17 (brittle culm3/*bc3* mutants) [59], and Y102 (knock down mutants of *Os4CL4*) [60], were used in this research. The seeds of rice genotypes were surface sterilized with a 10 percent sodium hypochlorite solution for thirty minutes. The seeds were washed with distilled water and incubated in a 30 °C chamber for two days in distilled water, replacing the water every 12 h. Germinated seeds were then shifted to Hogland’s solution and kept for two weeks. Later, the healthy seedlings of uniform height were transferred to rice fields at Guangxi University. Stem samples from the 2nd internode were collected at six growth stages. The characterization of growth stages was based on the 2nd internode length. Stage I to VI referred to the 2nd internode lengths of 0–2 cm, 3–5 cm, 6–8 cm, 10–12 cm, >13 cm, and mature internode, respectively. The samples were dried at 50 °C until a constant weight was achieved. The dried tissues were ground using a 40-mesh sieve and kept in a dry container until further usage.

### 4.6. Lignin Monomer Determination by HPLC and Statistical Analysis

The methods were described by Xu et al. [61] with minor modifications. Standard chemicals: p-hydroxybenzaldehyde (H), vanillin (G), and syringaldehyde (S) (Sinopharm Chemical Reagent Co., Ltd., Shanghai, China) were used for lignin monomer identification by HPLC. The samples were extracted from five genotypes at six growth stages with benzene-ethanol (2:1, *v*/*v*) in a Soxhlet for 4 h, and the remaining pellet (0.05 g) as cell wall residue (CWR) was added with 5 mL 2M NaOH and 0.5 mL nitrobenzene in the Teflon gasket with a stainless steel bomb. The bomb was tightly sealed and heated at 170 °C (oil bath) for 3.5 h and stirred at 20 rpm. The bomb was cooled with water, and the chromatographic internal standard (ethyl vanillin) was added to the oxidation mixture. This alkaline oxidation mixture was washed 3 times with 30 mL CH2C12/ethyl acetate mixture (1/1, *v*/*v*) to remove nitrobenzene and its reduction by-products [62]. The alkaline solution was acidified to pH 3.0–4.0 with 6 M HCl and was extracted with CH2CI2/ethyl acetate (3 × 30 mL) to obtain the lignin oxidation products in the organic phase. The organic extracts were evaporated to dryness under a reduced pressure of 40 °C, and the oxidation products were dissolved in 10 mL chromatographic pure methanol.

HPLC analysis: 20 μL solution was injected into HPLC (Waters 1525 HPLC) column Kromat Universal C18 (4.6 mm × 250 mm, 5 μm) operating at 28 °C with CH_3_OH: H_2_O: HAc (25:74:1, *v*/*v*/*v*) carrier liquid (flow rate: 1.1 mL/min). Calibration curves of all analytes routinely yielded correlation coefficients of 0.999 or better, and the detection of the compounds was carried out with a UV detector at 280 nm. The significant differences in lignin monomers and total lignins between NPB (WT) and four mutants were estimated using the Student *t*-test at *p*-values < 0.05 (significant *) and 0.01 (highly significant **). 

### 4.7. Real-Time PCR Assays of Twenty-Seven Lignin Biosynthesis Genes

To examine the transcriptome gene expressions, qRT-PCR analysis was used. The stem samples from 2nd internode were selected at five initial growth stages (I to V) as described above for each genotype. After collection, the stems were immediately frozen in liquid nitrogen and RNA was extracted using a Vazyme RNA isolation kit with three biological replicates. The genomic DNA was removed from RNA samples using Vazyme DNase I (Vazyme, Nanjing, Jiangsu, China). After purification, the Aidlab Truescript 1st Strand cDNA Synthesis Kit was used to synthesize cDNA. Real-time PCR was conducted using the PC59–2 x SYBR Green qPCR Mix (Aidlab Biotechnologies., Ltd., Beijing, China) in the DLAB accurate96 system and primers are listed in Appendix A. The qRT-PCR conditions were 95 °C for 2 min; 35 cycles of 95 °C for 15 s, 60 °C for 30 s; and 72 °C for 30 s. Three technical replicates from three biological replicates were used for each analysis, and the 2^−ΔΔCt^ method was used to determine the fold change in each gene. Rice *OsUBI* was used as an internal control for qPCR.

### 4.8. Calculation of Significant Association between Qpcr Gene Expression and Lignin Monomer Contents

The lignin monomer contents at five growth stages (dataset 1) (Appendix A) along with twenty-seven genes qPCR data (dataset 2) for each genotype (Appendix A) were fed to an online web server (https://www.bioinformatics.com.cn/basic_lncrna_mrna_pearson_spearman_coexpression_analysis_t013 (accessed on 10 September 2023)), which calculated the aggregated correlation coefficients and *p*-values between the two datasets. Meanwhile, some genes exhibited significant correlation coefficients, but very low gene expressions were ruled out as false positives. Finally, the significant correlation values were plotted in heatmap format between −1 and 1 in Tbtools. 

## 5. Conclusions

This study comprehensively analyzes lignin biosynthesis genes in rice, elucidating their phylogenetic relationships, expression patterns, and functional roles. The identification of 90 lignin biosynthesis genes and their classification into various families, such as PAL, C4H, 4CL, HCT, C3H, CCoAOMT, CCR, F5H, COMT, and CAD, underscores the complexity and diversity of lignin biosynthesis pathways. The significant contribution of duplication events to the expansion of these gene families highlights the evolutionary mechanisms driving lignin biosynthesis. Expression analysis across different tissues and developmental stages reveals distinct co-expression patterns, emphasizing the differential roles of these genes in primary and secondary cell wall formation. Through the analysis of lignin levels in the stems and the expression levels of genes unique to lignin biosynthesis in five selected genotypes, we discovered 10 potential genes associated with lignin composition and contents. These findings offer valuable insights into the genetic regulation of lignin biosynthesis and provide a foundation for future research to improve lignocellulosic biomass utilization.

## Figures and Tables

**Figure 1 ijms-25-10001-f001:**
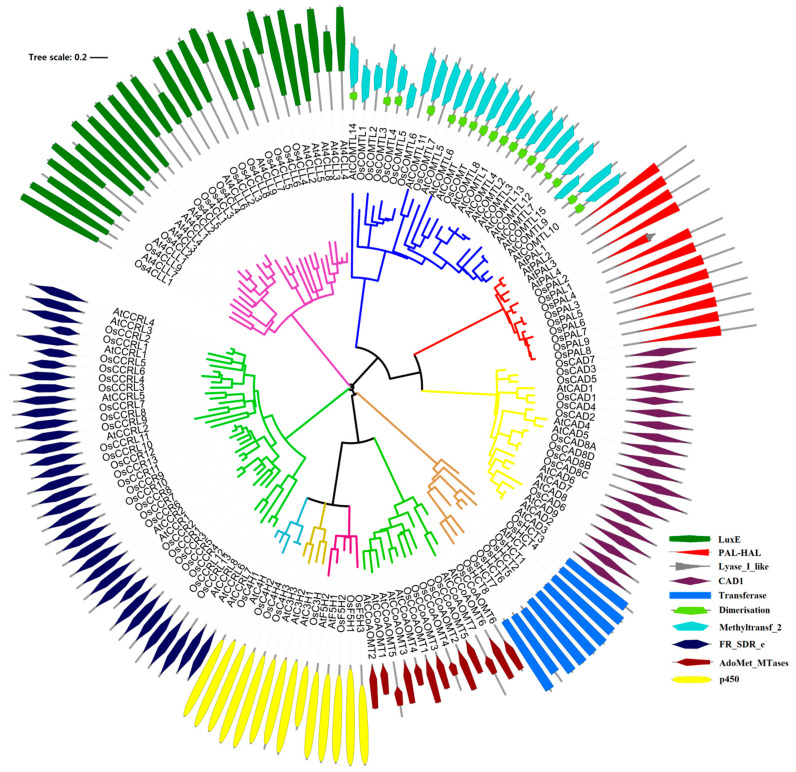
Unrooted phylogenetic tree of the rice and *Arabidopsis* lignin-related proteins. The unrooted phylogenetic tree of the rice and Arabidopsis was generated from the alignments of 9 OsPAL, 4 AtPAL; 4 OsC4H, 1 AtC4H; 14 Os4CL, 13 At4CL; 8 OsHCT, 1 AtHCT; 1 OsC3H, 3 AtC3H; 6 OsCCoAOMT, 7 AtCCoAOMT; 26 OsCCR, 10 AtCCR; 3 OsF5H, 2 AtF5H; 8 OsCOMT, 16 AtCOMT; and 11 OsCAD, 9 AtCAD protein sequences.

**Figure 2 ijms-25-10001-f002:**
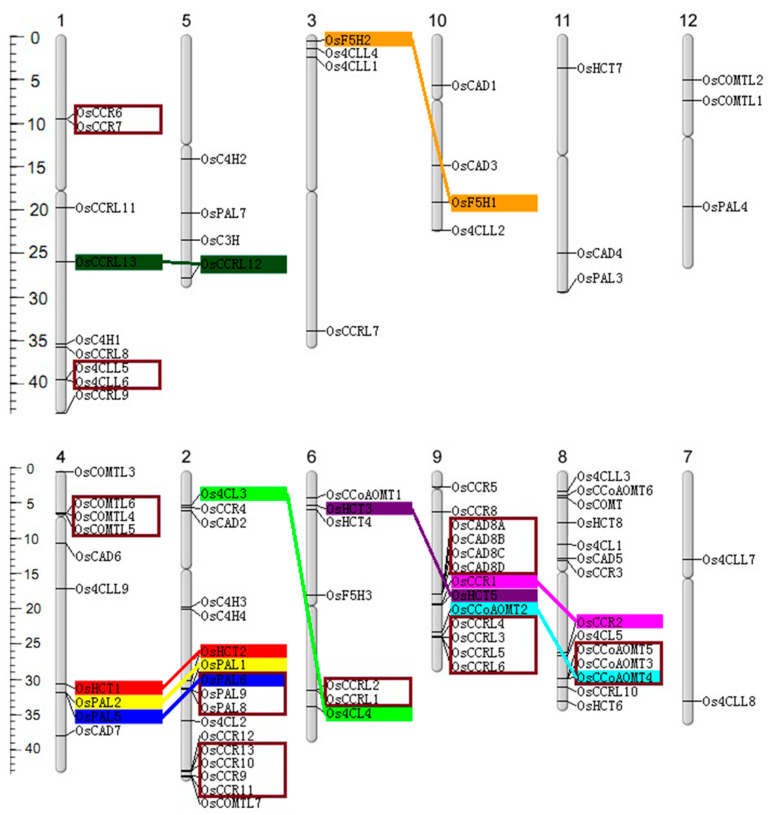
Chromosomal distribution, tandem, and segmental genome duplications of the lignin-related genes families in rice. The scale on the left is in megabases (Mb). The secondary constrictions on the chromosomes (vertical bars) indicate the positions of centromeres; the chromosome numbers are shown on the top of each bar. Gene names are placed on the right of chromosomes. The segmental duplications are highlighted with the same colors and connected with straight-colored lines. The tandem duplicated genes are enclosed crimson frame.

**Figure 3 ijms-25-10001-f003:**
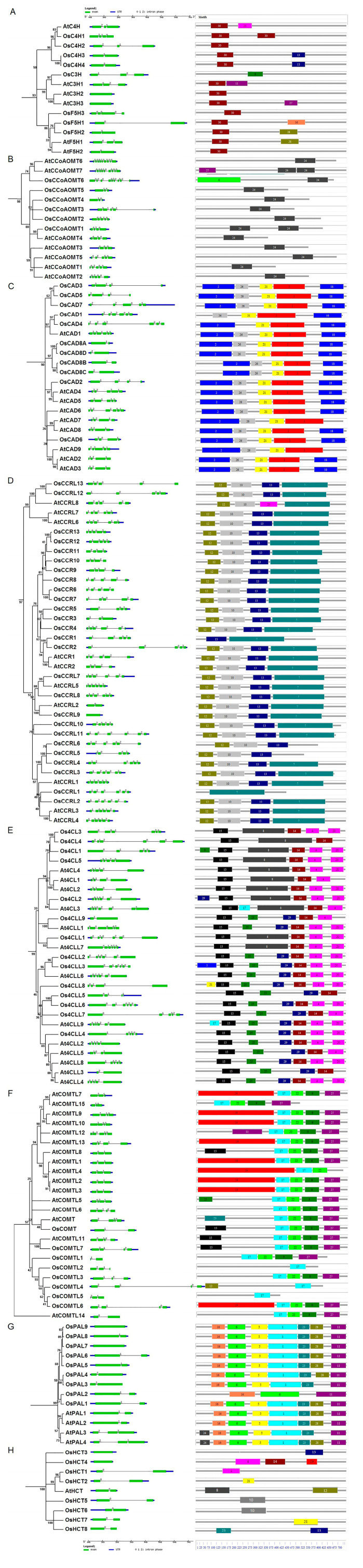
Gene structural organization and conserved protein motif analysis of lignin biosynthesis gene families in rice and Arabidopsis. (**A**) CH, (**B**) CCoAMT, (**C**) CAD, (**D**) CCR, (**E**) 4CL, (**F**) COMT, (**G**) PAL, and (**H**) HCT. The left side panel depicts gene structural organization, with exons represented in green boxes, introns in straight lines, and UTRs in blue boxes at the bottom. The panel on the right exhibits the conserved motifs, while each numbered box represents a single motif. The detailed motif sequences are listed in Appendix A.

**Figure 4 ijms-25-10001-f004:**
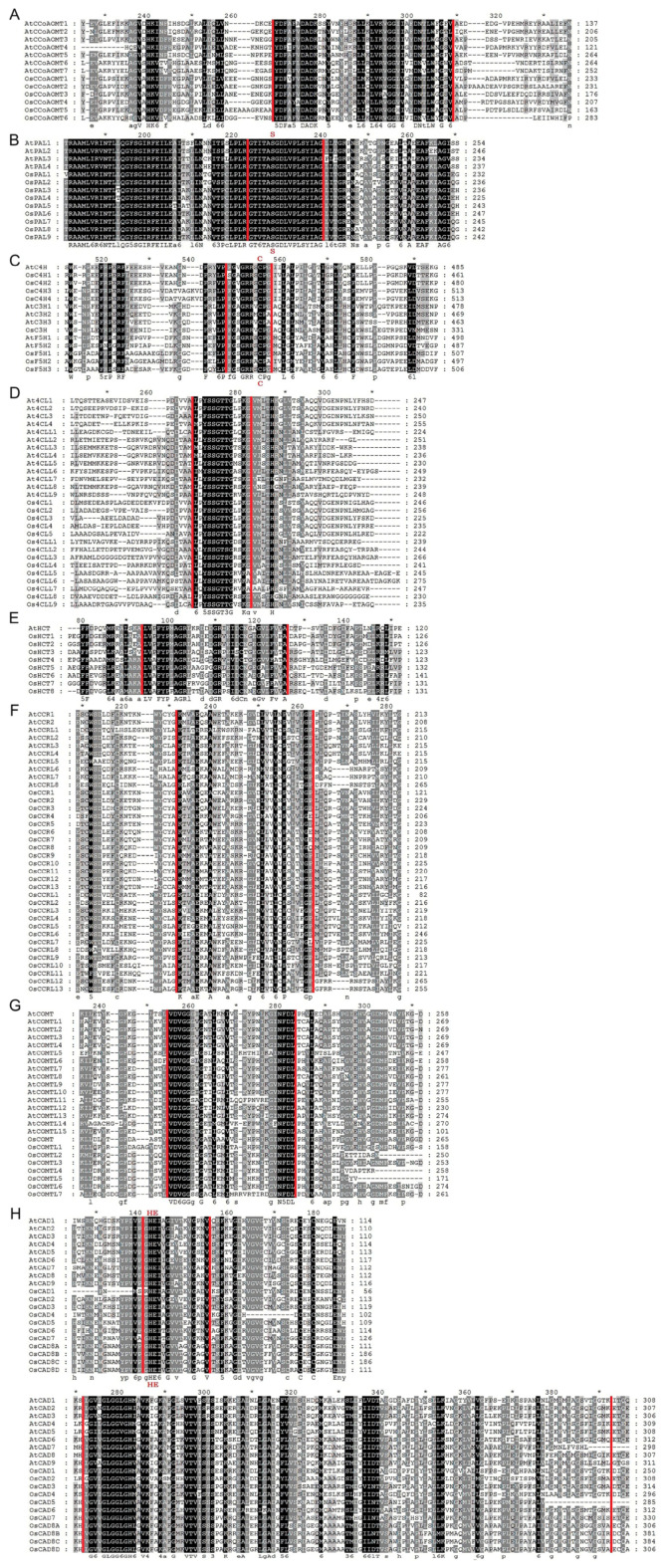
Multiple sequence alignment of lignin biosynthesis proteins. (**A**) CCoAOMT, (**B**) PAL, (**C**) C3H, C4H, and H25, (**D**) 4CL, (**E**) HCT, (**F**) CCR, (**G**) COMT, and (**H**) CAD. The darkness of shade is proportional to the level of conservation, such as residues highlighted in black indicate highly conserved, while those in grey are partially conserved or less conserved. Sequence regions within red vertical lines indicate the conserved motifs or domains. The star sign (*) indicates amino acids generally conserved in other contexts but exhibiting variations here.

**Figure 5 ijms-25-10001-f005:**
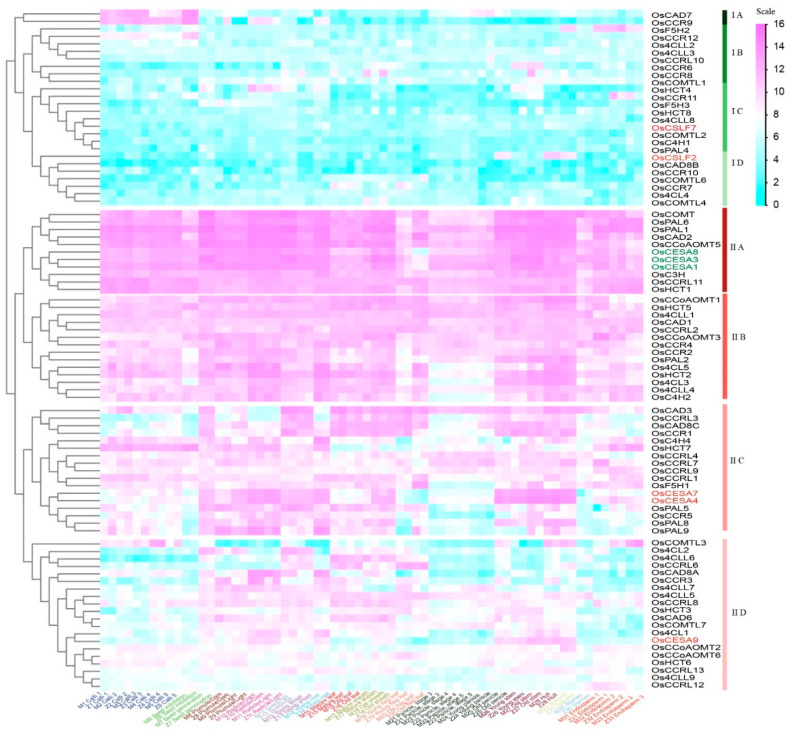
Hierarchical clustering of the gene expression of lignin families in the rice. The relative signal values of gene expressions are represented by a color scale on the extreme top right side. Blue signifies low expression, white indicates medium expression and magenta represents high expression. Genes belonging to the CESA gene family, like *OsCESA4*/*7*/*9*, and *OsCSLF2*/*7* are in brick red, while *OsCESA1*/*3*/*8* are in green, respectively, representing secondary and primary cell wall development markers.

**Figure 6 ijms-25-10001-f006:**
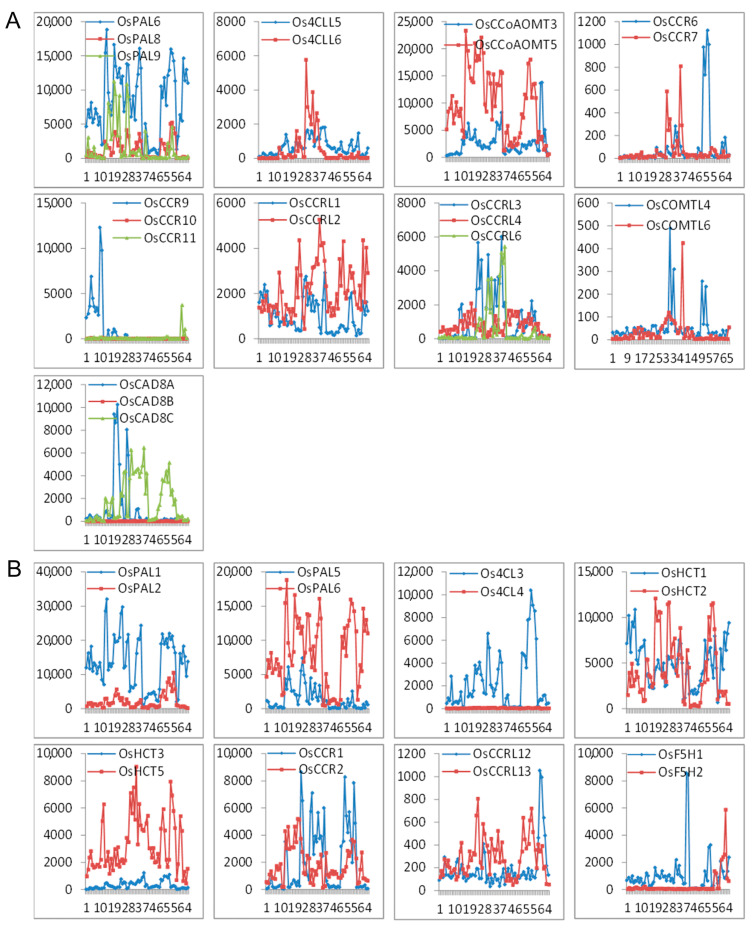
Expression patterns of the lignin-related genes as (**A**) tandem duplicates and (**B**) segmental duplicates. The *x*-axis represents the developmental stages. The *y*-axis represents the expression values obtained from the microarray analysis.

**Figure 7 ijms-25-10001-f007:**
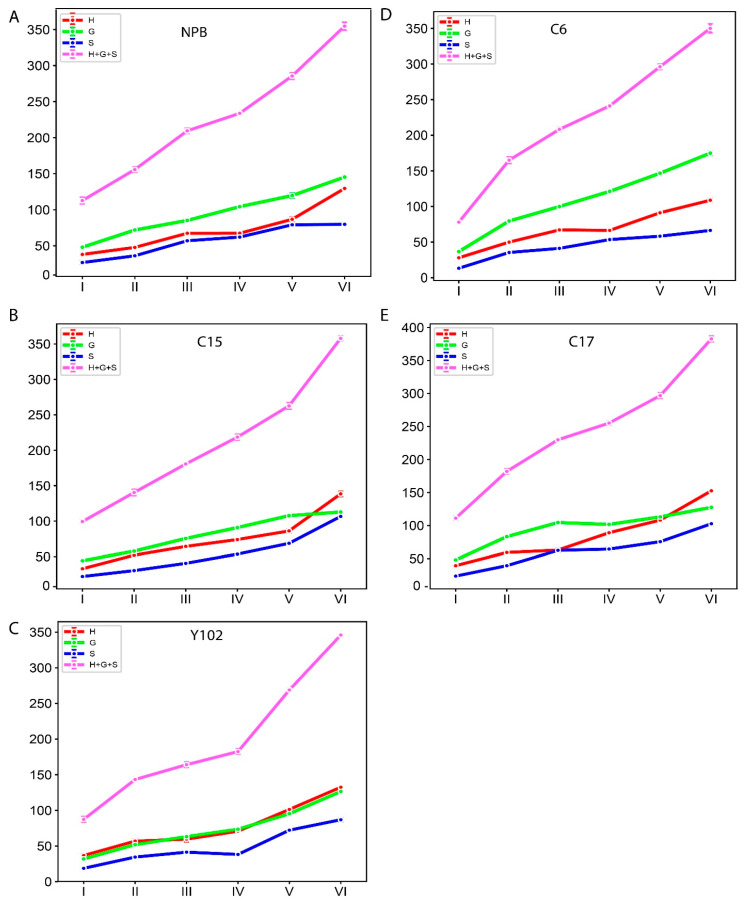
The monomer composition and total lignin contents (µmol/g dry matter) in six growth stages of rice stem in (**A**) NPB and mutants (**B**) C15, (**C**) Y102, (**D**) C6, and (**E**) C17. The six growth stages are depicted along the *x* axis (Stage I refers to the 2nd internode length is 0–2 cm; II refers to the length is 3–5 cm; III refers to the length is 6–8 cm; IV refers to the size is 10–12 cm; V refers to the length is greater than 13 cm; VI refers to mature internode). The monomer composition and total lignin of 2nd internode are exhibited along *y*-axis in “µmol/g dry matter”. Data are represented as means ± SD (*n* = 3).

**Figure 8 ijms-25-10001-f008:**
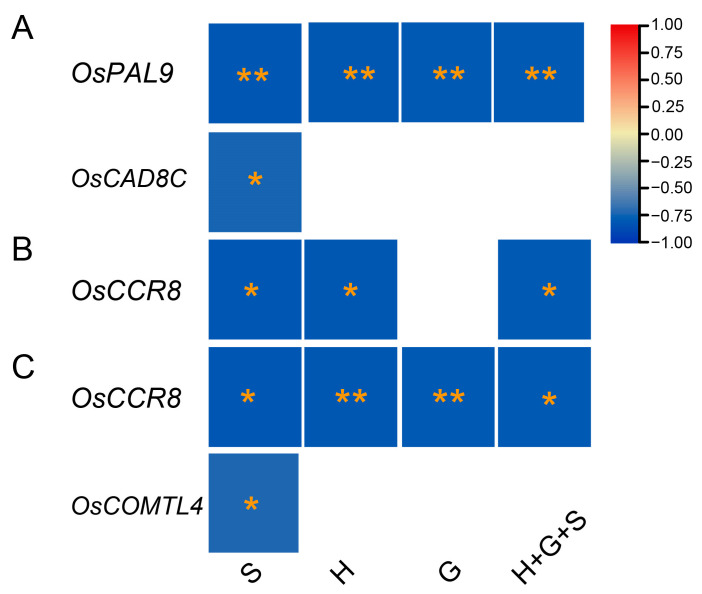
Significant negative associations between monomer contents and qPCR gene expressions in C6 (**A**), C17 (**B**), and Y102 (**C**) genotypes. The scale on the right indicates the strength and type of correlations, red refers to high positive correlations, while blue refers to high negative correlations. The significance of the correlation was calculated by *p* values (*p* < 0.01 = **, *p* < 0.05 = *) and denoted by yellow stars. Appendix A was used as the source of the lignin monomers and total lignins, while Appendix A were used as the source of gene expression data.

**Figure 9 ijms-25-10001-f009:**
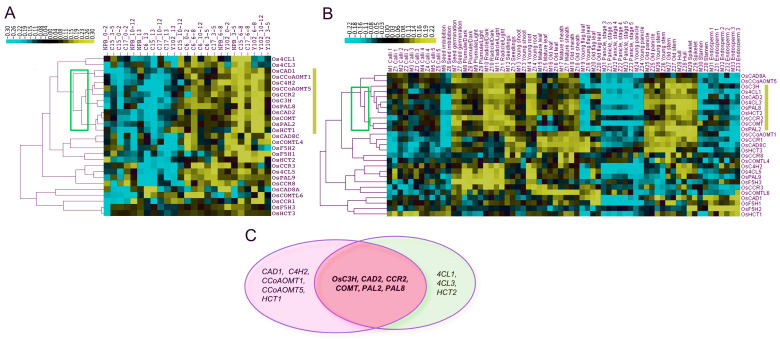
Identification of common co-expressed genes among qPCR and microarray expression datasets. (**A**) qPCR co-expressions (**B**) Micro-array co-expressions (**C**) overlap genes between both datasets. Eleven genes among twenty-seven genes in relation to the growth stage and genotypes. The eleven genes outlined in the yellow box indicate promising candidates. Five genotypes (NPB, C6, C15, C17, and Y102) and five periods (stI = 0–2 cm, stII = 3–5 cm, stIII = 6–8 cm, stIV = 10–12 cm, and stV > 13 cm) are represented as samples. The scale on the top left depicts the qPCR expressions as yellow for high, black for medium, and blue for low. The qPCR gene expression data are provided in Appendix A.

**Table 1 ijms-25-10001-t001:** The lignin-related genes co-expression comparison in rice and *Arabidopsis*.

	Rice			*Arabidopsis*	
Groups	Tissues	Genes	Groups	Tissues	Genes
Preferential expression in young vegetative tissues					
IA	Calli and seed imbibition	*Os4CLL2*; *OsCAD7*; *OsCCR9,10,12*; *OsHCT7*	/	/	/
IB	Calli and endosperm	*Os4CLL3*; *OsCCRL1,11*; *OsHCT1*	/	/	/
IIA	Radicle, young root, old stem, and hull	*Os4CL1,5*; *Os4CLL4,7*; *OsC4H2,4*; *OsCAD8A*; *OsCCR2,3,4,5*; *OsF5H3*; *OsHCT2,4*; *OsPAL9*	IIIC, IIID	Root	*At4CL1,2*; *AtC3H1*; *AtC4H*; *AtCAD4,6*; *AtCCoAOMT3,4,6*; *AtCCR1,2*; *AtCOMT*; *AtCOMTL1,2,3,8*; *AtF5H2*; *AtHCT*; *AtPAL1,2,4*
IIA, IIB	Seedlings, young shoot, old stem, and leaf	*Os4CL2,4*; *Os4CLL5,6*; *OsCAD3,6*; *OsCAD8B,8C*; *OsCCoAOMT6*; *OsCCR1,7*; *OsCCRL3,6,8*; *OsCOMTL1*; *OsHCT3*; *OsPAL5*	IID	Leaf and whole plant	*At4CLL9*; *AtCAD9*; *AtCCoAOMT7*
IIB	Old flag leaf, 14 days after heading	*Os4CLL1*; *OsCAD1*; *OsCCRL2*; *OsCOMTL6*; *OsF5H1*; *OsHCT5*	IIE, IIIC	Leaf	*At4CL1,2,5*; *AtC3H1*; *AtC4H*; *AtCCoAOMT3,4*; *AtCCR1*; *AtCCRL4*; *AtCOMT*; *AtCOMTL4*; *AtF5H1*; *AtHCT*; *AtPAL1,2,3,4*
IID	Endosperm	*OsC4H1*; *OsCCR11*; *OsCCRL9*; *OsCOMTL3*; *OsF5H2*	/	/	/
/	/	/	IIA	Seedlings and sepals	*At4CLL6*; *AtCAD7,8*; *AtCCoAOMT1*; *AtCOMTL5*
Preferential expression in reproductive stages					
IIA, IIB	Radicle, young root, old sheath, and old panicle	*Os4CL3*; *OsC3H*; *OsCAD2*; *OsCCoAOMT1,5*; *OsCCR6,8*; *OsCCRL13*; *OsCOMT*; *OsCOMTL4*; *OsHCT8*; *OsPAL1,2,6,8*	IIIA, IIIB, IIIC	Siliques and seeds	*At4CL1,2,4*; *At4CLL4,7*; *AtC3H1*; *AtC4H*; *AtCAD1,5*; *AtCCoAOMT3,4,5*; *AtCCR1*; *AtCCRL3,6*; *AtCOMT*; *AtCOMTL15*; *AtHCT*; *AtPAL1,2,4*
IIC	Stamen, one day before flowering	*Os4CLL8,9*; *OsCCoAOMT2,3*; *OsCCRL7,12*; *OsPAL4*	/	/	/
IID	Young panicle, heading stage	*OsCCRL4,10*; *OsCOMTL2,7*; *OsHCT6*			
/	/	/	IB	Flowers Stage 9, 10	*At4CLL1*; *AtCCoAOMT2*; *AtCCRL1*
/	/	/	IC	Flowers and siliques	*At4CL3*; *AtCCRL5,7,8*; *AtCOMTL11*
/	/	/	IIB	Flowers Stage 9, 10, 12	*AtC3H2,3*; *AtCAD3*; *AtCCRL2*
/	/	*/*	IIC	Flowers Stage 15 and stamen	*At4CLL8*; *AtCOMTL6,12,13*

Note. “/” indicates the missing tissue samples.

## Data Availability

Data is contained within the article and Appendix A.

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
