# Peer review of "Exploring Lignin Biosynthesis Genes in Rice: Evolution, Function, and Expression"

_ijms, 2024, doi:10.3390/ijms251810001_

Round 1

Reviewer 1 Report

Comments and Suggestions for Authors

In this paper, the genes related to lignin synthesis in rice were systematically analyzed, and six genes with important positions in lignin synthesis were found by combining the dynamic determination of lignin monomers and qRT-PCR gene expression analysis. The results are good helpful for exploring the formation and regulation of lignin in rice. The manuscript is rich in diagrams, while individual statements maybe to be verified.

 Line 158-159: The gene duplication analysis identified five pairs and four triplets of tandem duplicated genes, and eight pairs of segmentally duplicated genes. But from Figure 2, there 3 pairs, 3 triplets, 3 quadruple, and nine segmentally duplicated genes.

 Section 2.4: Where does the data for S, H, G, H+G+S in Figure 8? Is it derived from Table S3, S4? The source of the raw data in Figure 8 should be described.

Author Response

Comment 1: [Line 158-159: The gene duplication analysis identified five pairs and four triplets of tandem duplicated genes, and eight pairs of segmentally duplicated genes. But from Figure 2, there 3 pairs, 3 triplets, 3 quadruple, and nine segmentally duplicated genes.]

Response 1: [Thank you for pointing this out. We agree with this comment. Therefore, we have rewritten this part as the description in Figure 2 states. The revised manuscript change can be found on page 4, paragraph 1, and lines 123-125.]

Comment 2: [Section 2.4: Where does the data for S, H, G, H+G+S in Figure 8? Is it derived from Table S3, S4? The source of the raw data in Figure 8 should be described.]

Response 2: [Agree. We have accordingly revised, Table S3 as the source of H, G, S, and H+G+S data has been listed. The revised modifications can be found on page 19, lines 295-296, as the Figure 8 legends.]

Reviewer 2 Report

Comments and Suggestions for Authors

The manuscript presents molecular studies on the role of lignin-related genes in lignin biosynthesis in rice plants. On the base of five genotypes, several growth stages and tissue types the authors showed multiple correlations between the expression of a wide range of genes and the lignin content and lignin monomer composition.

The studies are important for understanding genetic control of the lignin biosynthesis, that  could help for example to improved bioenergy traits and lodging resistance in rice and other crops. The manuscript is well written. The way of presentation of used method and interpretation of results indicate high research competences of the team.

I do not feel expert to evaluate the correctness of the applied genetical and molecular methods in details, so my comments concern mostly rather more general issues:

“Materials and methods”

Comment:

Characterisation of plant cultivation method and details of sample collection as well as material preprocessing is lacking.

A short description of statistical approach could be also added.

Line 578-579

“Subsequent analysis of the gene expression data was performed in statistical computing language R (http://www.rproject.org) using packages available from the Bioconductor project (http://www.bioconductor.org).”

Comment: major R packages could be listed.

Line 324

“2.3. Metabolom Analysis of the lignin monomers in rice genotypes”

Comment: I suggest to remove the term “ Metabolom” since its meaning covers something different features  or  use “Biochemical” instead.

Figure 7.

Comment:

Error bars are surprisingly very low vs means, compared to reports in literature. Do they concern they really biological repetitions?

Table S3.

Title is rather non-adequate for the table content. Not only total lignin content is showed.

It is not clear which parameters are compared to each other and to which variables  the comparisons the “*” asterisk are referred? Student t-test can be applied only for comparison of two factors, is this the case in the study?

Table S4.

Meaning of (±) is not explained.

Author Response

Comment 1:[ “Materials and methods” Characterisation of plant cultivation method and details of sample collection as well as material preprocessing is lacking.]

Response 1: [Thank you for pointing this out. We agree with this comment. Therefore, we have added section 4.5 in the materials and methods about plant cultivation, sample collection, and pretreatments. The revised manuscript changes can be found on page 23, paragraph 4, and lines 461-472.]

Comment 2; [A short description of the statistical approach could also be added.]

Response 2: [Agree. A short description of statistical methods has been added. The added description of the statistical methods can be found on page 24, paragraph 2, and lines 490-492.]

Comment 3: [Line 578-579 “Subsequent analysis of the gene expression data was performed in statistical computing language R (http://www.rproject.org) using packages available from the Bioconductor project (http://www.bioconductor.org). Major R packages could be listed.”]

Response 3:[ Agree. The Pheatmap package was used in subsequent gene expression analysis. The addition of the R package name can be found on page 23, paragraph 3, line 459.]

Comment 4: [Line 324 “2.3. Metabolom Analysis of the lignin monomers in rice genotypes”. I suggest removing the term “ Metabolom” since its meaning covers something different features or using “Biochemical” instead.]

Response 4: [Agree. The term “metabolome” has been changed with the “biochemical” in the section 2.3 headings. The revised manuscript change can be found on page 17, paragraph 1, and line 256 of the revised manuscript.]

Comment 5: [ Figure 7. Error bars are surprisingly very low vs means, compared to reports in the literature. Are there concerns there are biological repetitions?]

Response 5: [As can be seen in Figure 7 error bars for total lignin's (H+G+S) are large compared to the individual lignin monomers, suggesting SE tends to increase with increasing mean values. There are biological repetitions.]

Comment 6: [Table S3. The title is rather inadequate for the table content. Not only total lignin content is shown.]

Response 6: [Agree. The title has been changed to “The lignin monomer composition and total lignin contents in five genotypes at six growth stages (µmol/g dry matter)”. The revised manuscript modifications can be found in the supplementary data word file on page 6, lines 36-37.].

Comment 7: [Table S3. It is not clear which parameters are compared to each other and to which variables the comparisons the “*” asterisk are referred. Student t-test can be applied only for comparison of two factors, is this the case in the study?]

Response 7: [Agree. NPB was used as the control group, while the four mutants were used as comparison groups. Therefore, the Student t-test was applied for the comparison of the two factors. The description of the control and comparison groups has been added in the main manuscript on page 23, paragraph 4, and lines 462-464 and Table S3 description in the supplementary data file on page 6, line 38.]

Comment 8: [Table S4. The meaning of (±) is not explained.]

Response 8: [Agree. The deviation around the mean values is indicated by the (±) sign. The desired manuscript modifications can be found in the Table S4 description in the supplementary data file on page 7, and lines 51-52.]